# A Study on the Current Impact on Island Tourism Development under COVID-19 Epidemic Environment and Infection Risk: A Case Study of Penghu

**Chien-Hung Wu**

Department of Marine Recreational, National Penghu University of Science and Technology, Penghu 880011, Taiwan; wu1023@gms.npu.edu.tw

**Abstract:** The present study examined the impact on island tourism development during the COVID-19 epidemic environment and infection risk by using Penghu as a case study. Using a mixed re-search methodology, 534 questionnaires were collected and analyzed using IBM SPSS Statistics 22.0 for Windows statistical software with statistical tests and t-tests. The views of scholars, experts, residents, and tourists on the questionnaire results were then compiled and finally examined by multivariate validation analysis. The results showed that different stakeholders maintained different perspectives on a number of economic, social, and environmental issues in the epidemic environment with risks of infection. Residents considered that the preservation of marine culture and the lack of resting and parking facilities for tourists are the issues that need to be improved in the development of Penghu tourism. Visitors believe that improving littering, vessel mooring space, pollution from heavy oil discharges, landscape and historic site protection, surface litter and pollution in the harbor, marine habitat, heavy oil spills, tourist litter, and threats from invasive species will help attract tourists to visit and spend money.

**Keywords:** COVID-19; island tourism; yacht tourism; oil pollution; environmental damage

## 1. Introduction

Tourism is a human activity and an industrial model that is driven by the natural beauty and diverse ecosystems of the local area [1] as well as the attractiveness of the existing cultural and other related tourism resources [2]. Among them, due to the island terrain surrounded by the sea, diverse marine ecology, as well as the characteristics of natural geological landscape and settlement culture, island tourism has gradually become popular with tourists [3].

The Penghu Islands are located to the south of the Taiwan Strait. Due to the mutual compression of the earth's crust and volcanic eruption, 96 islands of different sizes were formed, resulting in a coastline of 448.974 kilometers, and the terrain descends from south to north. Due to geological and geographical environmental factors, the local area has basalt landscapes, characteristic islands with various appearances, and multiple marine ecological resources, such as green turtle conservation areas. In addition, there are special humanities, food, art, customs, history, and architectural features, such as the very historic Mazu Temple and the Ocean and Geological Museum [4]. There are also activities, such as the heavy sailing tourism industry promoted in the later period and the firework festival that combines scenery and technology [5], as shown in Figure 1. As a result, tourism industries and activities, such as shipping, festivals, arts, humanities, and history, have been created, which have attracted a large number of visitors [6] and have created countless business opportunities for local villages and residents in the Penghu Islands. Penghu is committed to developing tourism activities through relevant resources. Before the outbreak of the epidemic, the number of tourists reached a maximum of 1,286,000, with

11,100 service-related establishments, 33,200 jobs, and a total production of 191.69 million USD [7]. This shows the effectiveness of tourism development in the region.

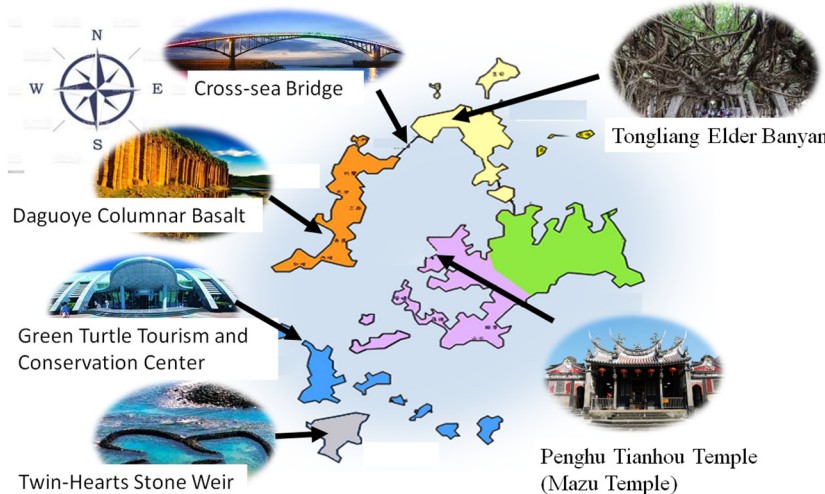

**Figure 1.** Overview of famous tourist attractions in Penghu.

Since the outbreak of COVID-19 in December 2019, national and international tourism activities around the world have been suspended due to the risk of infection by the epidemic [8], making 2020 arguably the most difficult year for the global tourism industry [9,10]. The outbreak has raised concerns about the risk of infection while affecting the local tourism and economic development of Penghu [11], which is a dilemma. Before the outbreak, tourism provided a great deal of business opportunities and economic income, but the negative environmental impacts of tourism development, such as tourism noise, garbage, and waste oil pollution, troubled residents [1]. The COVID-19 outbreak resulted in less than 10,000 visitors to Penghu each month. It is estimated that the total number of visitors will be less than 1 million in 2021. The current unemployment rate has reached 4.8%, which is the highest in 10 years [11,12]. It is evident that the epidemic has dealt a serious blow to the development of tourism and the livelihood of the residents in Penghu. In addition, tourism development may have an impact on local development due to management decisions and implementation effectiveness [1,11], which will affect the local economy, society, and environment [11–14] and act as an aid or hindrance to sustainable development.

Scholars have suggested that although the epidemic has changed the pattern of human life and severely affected tourism [8–14], policy making is the best way to address the local plight [13]. Tourism impact theory is the best model for exploring the changes that occur after the promotion of tourism decisions [15–17]. Therefore, using tourism impact theory to explore the present situation would be beneficial to understand the extent to which the epidemic has affected the current state of tourism development on the island. Verification of the impact of decisions on the environment takes time [14], whereas the impact of tourism activities can be verified only after the activity is over [1,15–17]. Therefore, people's perceptions can provide accurate information when exploring the changes that occur in places due to the implementation or promotion of decisions or activities [18–20].

Scholars believe that long-term residents of the area can detect subtle changes in their surroundings because they are always in contact with the environment [1,16], and therefore, the impact of development can be identified through the perspective and experience of the residents [16,20]. On the other hand, since tourism development mainly provides outside visitors with the experience of a tourist destination [14,21], an objective view can be obtained by exploring the changes and the effectiveness of promoting tourism through the perspective and feelings of tourists [21–23]. However, both residents and visitors satisfy their needs by exploiting local natural resources [1,14,21], and the quality of life of residents

and the experience of visitors will be influenced by the quality of development decisions. It is clear that both are influenced to some extent by the effects of development. Therefore, obtaining the opinions of residents and visitors and finding a balance will help to find more detailed results.

The sudden impact of the outbreak has had a serious impact on the global travel industry. Before a vaccine can completely stop the virus, government agencies and scholars around the world are trying to find solutions to mitigate or ameliorate the negative effects of the virus [24–26]. According to the literature review, in recent years, scholars have explored many different issues regarding the impact of the epidemic on tourist areas [10–15,27,28], including investigating the impact of tourism in island areas [29,30] and studying tourism in the Penghu Islands [31]. However, the impact of the epidemic on the tourism industry of the Penghu Islands has not been investigated.

Therefore, we believe that analyzing the impact of the epidemic on the development of tourism in the islands using the Penghu Islands as a case study will help us understand the current situation of the local tourism industry in the Penghu Islands, formulate future countermeasures, and improve the shortcomings of island tourism research.

In summary, this study analyzes the impact of the epidemic on island tourism development from the perspectives of residents and tourists, using Penghu as an example. The purpose of this study is to propose countermeasures to the current situation of tourism in the Penghu Islands in order to improve the shortcomings of government tourism policies and island tourism research in the region and to achieve the goal of sustainable tourism development under the epidemic.

## 2. Literature Discussion

### 2.1. Paradox Generated by Tourism Development—Tourism Sustainability and Overtourism

Tourism is a recognized way to stimulate local economic industries and development [1–3]. For island countries or regions that are not rich in minerals and resources and do not have highly sophisticated industries and manufacturing sectors, using tourism industry and resources to promote economic growth and improve local development is the most common means [1,3,5].

However, over the years, and as research studies have shown, tourism development is no longer the green economy that it is claimed to be. Studies have confirmed that although tourism development can bring positive economic benefits, improve community environment, and raise people's living standards [1–3,13,20], over-promotion of tourism activities and attraction of people can lead to the loss of local cultural characteristics, destruction of villages and ecological environment, and a sharp increase in local consumption costs and a decline in quality of life [1–3,13,20,32]. When tourism development is not satisfactory or even stagnant, although the villages and environment of the island area get a respite, the instantaneous disappearance of economic income and sources will lead to a threatened livelihood for the residents and local businessmen [33]. It is evident that the development of tourism cannot be based on mere promotion. In order to stimulate economic development, a proper and rational vision and planning is needed to achieve a balance.

Sustainable tourism development is not just a slogan; it is a concept and framework. Tourism policy development mainly considers how to improve the local development difficulties, enhance the village economy, make the industry development flourish, and improve the quality of life of the people [1,3,5]. However, at present, more attention is paid to the degree of maintenance of the local natural and ecological environment, whether local culture and historical buildings can be preserved, whether residents' physical and mental health can be protected, and whether local tourism resources and development can achieve the goal of survival, coexistence, and mutual prosperity [1–3,13,20,32]. Therefore, researchers believe that the current tourism development on the island should not only acquire the natural environment and local industrial resources in order to promote local economic and human social prosperity. The goal of preserving tourism development resources; promoting the survival, coexistence, and mutual prosperity of natural ecology

and human beings; and achieving the goal of sustainable resources are the appropriate actions and goals for tourism development on islands under the epidemic.

### 2.2. The Importance of the Perceptions of Different Stakeholders in Island Tourism Development in an Epidemic Environment

The impact of the epidemic on tourism has been significant [24–26,34–36], but island villages with rich marine resources still need industries such as tourism to promote development. Thus, even in a severe epidemic environment, island villages and residents still expect the economy of their communities and the quality and conditions of life of individuals to improve as a result of tourism development [22,23]. It is evident that the effectiveness of tourism development is an important issue for the residents.

In addition, the epidemic has made tourism very difficult [24–26,36], but tourists still expect to experience beautiful natural environmental resources to improve their physical and mental health and hope that the local cultural and marine resources feature a pleasant experience [1,11]. It is evident that tourists are also expecting tourism development to be effective.

Policies can bring changes to a place. Villages expect tourism development to bring economic, social, and environmental improvements to their places [16,17]. However, residents expect to improve their personal living environment and conditions [16,20], while tourists expect to enhance their physical and mental status through tourism experiences [14,21]. Both use the available resources to meet their respective needs, but the conditions for satisfaction are different [1]. Therefore, scholars believe that it is better to obtain the opinions of both sides, explore the current situation of local tourism development, and find ways of improvement to satisfy the needs of both sides as much as possible, and the final solution will be better [1,14,34].

### 2.3. Economic Impact on Island Villages

Tourism development is mainly expected to bring economic benefits to the local community by attracting tourists to visit and spend money through the appeal of local tourism resources [1,16]. Therefore, the change of village economic development is arguably the most direct and priority impact of tourism development.

The impact of tourism development on village economic development can be explored at the level of consumer prices, industrial construction, and community development, according to scholars [31–34]. Among them, issues on local employment, wages, incentives, consumption, leisure activities, industry, construction, facilities, prices, health, culture and creativity, community feedback, and policy coordination [1,14,34,35] will be the main influences.

Hence, the analysis of the current state of economic development in terms of local employment, industry, facilities, prices, wages, consumption, construction, incentives, culture and creativity, leisure activities, health, community feedback, and policy coordination can provide an understanding of the impact of tourism development in the island on the local economic level in the context of the epidemic.

### 2.4. Social Impact on Island Villages

Due to geographical constraints, traditional island villages rely on marine resources for their livelihoods and economic resources, generating unique architectural and cultural attractions with tourism value [14,15]. However, as the tourism industry advances and develops, the habits and culture of established communities will change to some extent [14,16]. Therefore, the social development changes in villages are also influential factors under tourism development.

It is suggested by scholars that the changes brought about by tourism development on the social aspect of villages can be discussed in terms of community creation, living atmosphere, and cultural safety [1,37]. Among them, the quality of tourism services and activities, commercialization of community buildings, community development participation, tourism indicators, recreational facilities, community environment, indigenous

culture, vocational training opportunities, living environment, youth development, public interaction, preservation of traditional culture, community security and safety, investment in cultural industries, and desire to revisit or the idea of settling down in the island may also be altered [1,38,39].

Therefore, an analysis of the current state of social development in terms of the quality of tourism services and activities, commercialization of community buildings, community development participation, tourism indicators, recreational facilities, community environment, indigenous culture, vocational training opportunities, living environment, youth development, public interaction, preservation of traditional culture, community security and safety, investment in cultural industries, desire to revisit or settle down in the island, etc., can provide an understanding of the impact of tourism development on the social aspects of the island under the epidemic.

### 2.5. Environmental Impact on Island Villages

The island has become a unique tourism environment and resource due to the geographical advantage of being surrounded by the sea, the diversity of marine ecology, and the architectural features of the local fishing villages [14,15,31]. However, with the promotion of the tourism industry and development, the existing village communities and the surrounding marine environment will be changed to a certain extent [14,16,40]. Therefore, it is inevitable that the existing village environment will be affected by tourism development.

Scholars point out that promoting tourism development requires the use of local resources, such as natural ecology and space, so the existing environment may change, especially the natural and community environments that have the greatest impact [23,41]. Among them, the community environment will be affected in terms of transportation facilities, public transportation for tourism, management of bicycle paths and trails, bicycle rental, Wi-Fi network coverage, management manpower, living space, distribution of tourism waste, tourism building area, parking, and resting facilities [14,16,23,41] as well as in terms of air quality, emissions, water quality, native habitats, vegetation and woodlands, and environmental awareness in the natural environment [1,38,39].

Therefore, by examining the environmental issues in terms of transportation facilities, public transportation for tourism, bicycle and tourist trail management, bicycle rental, Wi-Fi network coverage, management manpower, living space, distribution of tourist waste, tourism building area, parking, and resting facilities and the natural environment in terms of air quality, exhaust emissions, water quality, native habitat, vegetation and woodland, and environmental awareness, we can understand the impact of tourism development on the local environmental aspects of the island under the epidemic environment.

## 3. Methods

### 3.1. Research Process and Framework

Although the global tourism industry is being affected by the COVID-19 outbreak [7–9], ameliorating the local plight is the goal of governmental decisions [1,13]. However, decisions and developments can still have both positive and negative effects. Therefore, we analyzed the current situation of tourism development in the island during the COVID-19 epidemic environment with risks of infection in terms of economic [1,11,29,30], social [1,38,39], and environmental [14,16,23,41] aspects and from the perspective of different stakeholders by referring to the results of national studies in the literature and the theoretical framework of related studies [1,14,21]. The research framework is shown in Figure 2.

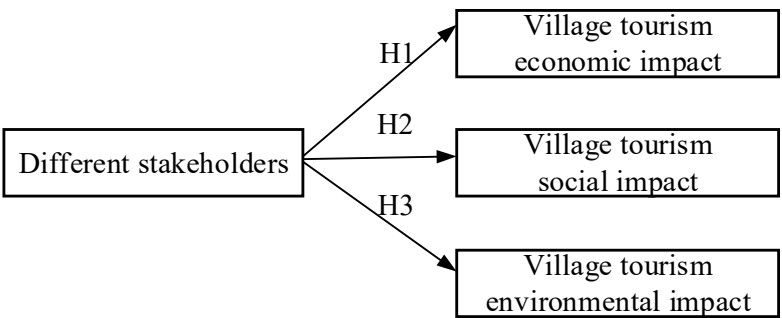

**Figure 2.** Research framework.

*3.2. Research Hypothesis*

Based on the above description, the hypotheses were:

**Hypotheses 1 (H1).** *Different stakeholders share the same perceptions of the impact of economic development on island villages.*

**Hypotheses 2 (H2).** *Different stakeholders share the same perceptions of the impact of social development on island villages.*

**Hypotheses 3 (H3).** *Different stakeholders share the same perceptions of the impact of environmental development on island villages.*

*3.3. Research Tools*

The purpose of this study was to analyze the current impact of tourism development on the island during the COVID-19 epidemic environment with risks of infection from the perspective of different stakeholders. The questionnaire was divided into 2 parts. The first part provided background information on gender (male, female), age (under 20, 21-29, 30-39, 41+), and identity characteristics (resident, tourist) of the sample. The second part examined the current status of village impacts, which were further divided into economic, social, and environmental dimensions based on the results of the literature analysis. Fourteen questions were designed in the economic dimension with reference to [1,14–17,37–39]; 11 questions in the social dimension with reference to [11–13,28,37]; and 15 questions in the environmental dimension with reference to [14,17,23,39,41].

Except for the background information, a 5-point Likert-type scale was used for all issues (1 represents strong disagreement and 5 represents strong agreement). The questionnaire was first compiled by referring to tourism-related literature. Then, 100 questionnaires were distributed in March 2021, which were analyzed and validated using IBM SPSS Statistics 22.0 for Windows statistical software. The scale was suitable for continuous factor analysis when the Kaiser-Meyer-Olkin (KMO) value > 0.06 and the p-value in Bartlett's test was less than 0.01 ($p < 0.01$) [40]. When the coefficient $\alpha$ was greater than 0.60 it indicated that the reliability of the questionnaire was good [40] and could become a formal question for further investigation. The results of the analysis are shown in Table 1.

Based on the literature [1,14–16,36–38], a 14-question economic dimension questionnaire was developed. The analysis revealed that it had a KMO value > 0.919, a Bartlett's approximate $\chi^2$ value of 4848.841, a degree of freedom (df) of 91, and a significance of 0.000 ($p < 0.001$), making it suitable for factor analysis. The explained variance of the scales was 27.17%, 21.528%, and 12.366%, respectively, and the total explained variance was 61.065%. All questions were subjected to factor analysis and were retained in order to understand the actual state of economic development. The economic facet are divided into three aspects: industrial development (Design 4 questions), employment and consumption (Design 4 questions), and community development (Design 6 questions), with a total of 14 questions (economic). The Cronbach's alpha for the three scales was 0.927, 0.925, and 0.933, respectively, which indicated that the scales had high reliability and could be used for further study.

**Table 1.** Analysis of the current island tourism development impact questionnaire.

| Facets | Secondary Facets | Issues | Cronbach's $\alpha$ |
|---|---|---|---|
| Economic impact of island villages | Employment and consumption | Improve job opportunities Enhancing entrepreneurship opportunities for maritime activities Increase work wages and income Increased costs and consumer spending | 0.926–0.927 |
| | Industrial development | Increase in the number of tourism construction and industries Matching local characteristic industry cooperation Increase leisure opportunities Improve the quality of public facilities and maintenance | 0.924–0.925 |
| | Community development | Benefits of tourism development to give back to the community Improve the accessibility of public transportation Land, house prices and prices are rising Improve hygiene quality Formulate development protection policies Develop cultural and creative products | 0.926–0.933 |
| Social impact of island villages | Village construction | Increased tourism visibility Increase the quality of village tourism services and activities Add travel activity boards or indicators, billboards Increase the construction of recreational facilities in the village Promote Yacht Development Organization | 0.923–0.924 |
| | Quality of life | Facilitate the development of local youth returning to their hometowns Improve the protection of marine culture Increase the protection of coastal style buildings | 0.924–0.926 |
| | Culture and safety | Harmonious interaction between tourists and residents Reduce the burden on police, fire and security personnel Willingness to revisit or buy property on the island | 0.923–0.934 |

| Facets | Secondary Facets | Issues | Cronbach's $\alpha$ |
|---|---|---|---|
| Environmental impact of island villages | Community environment | The problem of tourists littering has increased<br>Ship berthing space is still insufficient<br>Tourist rest and parking facilities are still insufficient<br>Tourists are the cause of damage to the environmental quality of tourist villages<br>Increase pollution caused by heavy oil emissions<br>Increase in the amount of tourism waste<br>Historic sites restoration | 0.936–0.943 |
| | Ecological environment | Increase in garbage in ports and on the sea<br>Increased port water pollution<br>Excessive development of the coast or land around the port<br>There are enough public toilets around the port<br>The destruction of marine habitats<br>Marine pollution caused by heavy oil discharge<br>Increased tourism garbage<br>Alien species threaten local ecology | 0.935–0.937 |

Based on the literature [14–16,30,39], an 11-question social dimension questionnaire was developed. The analysis revealed that it had a KMO value > 0.932, a Bartlett's approximate $\chi^2$ value of 4107.088, a degree of freedom (df) of 55, and a significance of 0.000 ($p < 0.001$), making it suitable for factor analysis. All questions were subjected to factor analysis and were retained in order to understand the actual state of economic development. The explained variance of the scales was 30.408%, 24.785%, and 12.040%, respectively, and the total explained variance was 67.233%. All questions were subjected to factor analysis and were retained in order to understand the actual state of economic development. The social facet are divided into three aspects: village construction (Design 5 questions), culture and safety (Design 3 questions), and quality of life (Design 3 questions), with a total of 11 questions (social). The Cronbach's $\alpha$ for the three scales were 0.924, 0.926, and 0.934, respectively, which indicated that the scales had high reliability and could be used for further study.

Based on the literature [14,16,23,38,40], a 15-question environmental dimension questionnaire was developed. The analysis revealed that it had a KMO value > 0.925, a Bartlett's approximate $\chi^2$ value of 6644.946, a degree of freedom (df) of 105, and a significance of 0.000 ($p < 0.001$), making it suitable for factor analysis. All questions were subjected to factor analysis and were retained in order to understand the actual state of economic development. The environmental facet are divided into three aspects: ecological environment (Design 8 questions) and community environment (Design 7 questions), with a total of 15 questions (environmental). The Cronbach's alpha for the two scales was 0.943 and 0.947, respectively, which indicated that the scales had high reliability and could be used for further study.

All local residents living in Penghu and tourists visiting the area were the target population of this study. From March to April 2020, the team used the snowball sampling method to collect tourist samples at various scenic spots in Penghu as well as airports and

ports in other places. At the same time, field surveys were conducted locally, and samples of residents were collected simultaneously. However, due to the consideration of the epidemic, the online questionnaire platform was also used to collect sample information. The team finally obtained 543 valid questionnaires and used IBM SPSS Statistics 22.0 for Windows statistical software to analyze the questionnaire data with the basic statistical verification and the *t*-test method.

Next, six respondents, including residents, experts and scholars, and businesses owners and tourists, were invited to present their views based on the data results, as shown in Table 2. We then used multivariate verification analysis to first compile and categorize the data in a sequential manner and then compared and corroborated the information to obtain valid information to explore the current situation of the impact of tourism development on the island during the COVID-19 epidemic environment and infection risk.

**Table 2.** Respondent's background information and an overview of the interview outline.

| Identity | Gender | Residence Time/Years of Work Experience | Identity | Gender | Residence Time/Years of Work Experience |
|---|---|---|---|---|---|
| Resident | Male | 25 | Tourist | Male | 9 |
| Resident | Female | 30 | Tourist | Female | 5 |
| Professor | Male | 15 | Entrepreneur | Female | 15 |
| **Construct** | **Issues** | | | | |
| Impact of tourism development | 1. What are the impacts on the tourism development of local villages during the epidemic? Please briefly explain the reasons and suggest improvements. 2. What is the impact on the economic, social, and environmental conditions of the village? Please briefly explain the reasons and suggest improvements. | | | | |

### 3.4. Methodology Analysis and Limitations

A mixed research design involves obtaining public perceptions through a data-based survey and then obtaining representative information through a qualitative survey. Comparing and validating data from both sources can fill research gaps where theories or questions have not been explored [42,43]. This study adopted a mixed research approach to investigate the perceptions of local residents and tourists visiting Penghu Island on the impact of the current situation of local tourism development while living and traveling during the epidemic outbreak. The initial phase of the study began in March 2021, but due to the large size of the study site and the human, material, and financial constraints of the research team, an adequate sample size could not be collected in the field. Moreover, although the information was collected in combination with an online questionnaire platform, the respondents were already limited by the screening criteria first. From December 2019, the COVID-19 outbreak has not yet subsided, thus limiting the number of samples collected by the researchers. The shortcomings of this study will be presented at the end of the paper with suggestions for improvement in anticipation of refinement by subsequent researchers.

The researcher clearly stated the significance and objectives of the interview questions before conducting the sample collection. The questionnaires and interview questions were clearly written to describe the purpose of the study. All information was obtained for analysis with the consent of the respondents after asking for their consent again at the end of the sampling or interview. Therefore, all analyzed data were authorized by the respondents and conformed to the guidelines.

### 4. Result Analysis

### 4.1. Sample Background Analysis

Analysis of the background information of the sample of 534 questionnaires obtained by the study showed that there was not much difference in the number of gender and identity. The majority of the sample were male (299; 56%), and residents (291; 54.49%);

most of them were aged 21–30 (288; 47.8%), and the least aged 41 (73; 13.7%). As shown in Table 3.

**Table 3.** Sample background analysis.

|  |  | N | % |
|---|---|---|---|
| **Gender** | Male | 299 | 56% |
|  | Female | 235 | 44% |
| **Identity** | Resident | 291 | 54.49% |
|  | Tourist | 243 | 45.51% |
| **Age** | Under 20 | 97 | 18.2% |
|  | 21–30 | 255 | 47.8% |
|  | 31–40 | 109 | 20.4% |
|  | Above 41 | 73 | 13.7% |

*4.2. Analysis of the Perceptions of Different Stakeholders on the Economic Impact of Tourism in Island Villages during the Epidemic*

According to the literature, tourism development can alleviate local economic distress and promote business opportunities [1,16]. However, the analysis revealed significant differences ($p < 0.05$) for the issues of increased job opportunities (4.01:4.00), increased job salary income (4.04:4.00), and benefits of tourism development clearly returning to the community (3.07:3.93), while no significant differences were found for other issues. Evidently, different stakeholders believed that under the current epidemic environment, there were different views on the issues of increased job opportunities, increased job salary income, and benefits of tourism development clearly returning to the community in the local economy and that residents felt more strongly about the changes in the current situation of job opportunities and job salary income, while tourists felt more strongly about the benefits of tourism development clearly returning to the community. This result is inconsistent with hypothesis 1 of the study. As shown in Table 4.

**Table 4.** Analysis of the perceived economic impact of tourism in island villages.

| Secondary Facets | Issues | Resident | | Tourist | | *p* |
|---|---|---|---|---|---|---|
|  |  | M | SD | M | SD |  |
| Employment and consumption | Improve job opportunities | 4.02 | 0.840 | 4.00 | 0.651 | 0.010 * |
|  | Enhancing entrepreneurship opportunities for maritime activities | 4.10 | 0.725 | 4.15 | 0.691 | 0.246 |
|  | Increase work wages and income | 4.04 | 0.769 | 4.00 | 0.673 | 0.012 * |
|  | Increased costs and consumer spending | 3.86 | 0.843 | 3.88 | 0.771 | 0.096 |
| Industrial development | Increase in the number of tourism construction and industries | 4.10 | 0.707 | 4.31 | 0.634 | 0.124 |
|  | Matching local characteristic industry cooperation | 4.21 | 0.696 | 4.26 | 0.683 | 0.358 |
|  | Increase leisure opportunities | 4.35 | 0.683 | 4.23 | 0.674 | 0.862 |
|  | Improve the quality of public facilities and maintenance | 3.86 | 0.829 | 3.95 | 0.794 | 0.172 |
| Community development | Benefits of tourism development to give back to the community | 3.75 | 0.859 | 3.93 | 0.768 | 0.004 * |
|  | Improve the accessibility of public transportation | 3.62 | 0.987 | 3.76 | 0.878 | 0.094 |
|  | Land, house prices and prices are rising | 3.59 | 0.914 | 3.69 | 0.872 | 0.672 |
|  | Improve hygiene quality | 3.49 | 0.909 | 3.58 | 0.858 | 0.613 |
|  | Formulate development protection policies | 3.66 | 0.858 | 3.86 | 0.863 | 0.127 |
|  | Develop cultural and creative products | 3.90 | 0.813 | 4.05 | 0.757 | 0.481 |

* $p < 0.05$.

### 4.3. Analysis of the Perceptions of Different Stakeholders on the Social Impact of Tourism in Island Villages during the Epidemic

According to the literature, when promoting tourism development, policymakers will invest in improving the tourism environment and health conditions of local communities to meet the needs of tourists but also to create conflicts with established culture and lifestyle [14–16]. The analysis revealed no significant differences ($p < 0.05$) except for the issue of promoting the preservation of marine culture (4.07:4.00). It is evident that different stakeholders have different views on the issue of promoting the preservation of marine culture at the social level of local tourism development in the island and that residents have higher perceptions of the effectiveness of marine culture preservation. This result is consistent with research hypothesis 2. As shown in Table 5.

**Table 5.** Analysis of the perceived social impact of tourism in island villages.

| Secondary Facets | Issues | Resident | | Tourist | | $p$ |
|---|---|---|---|---|---|---|
| | | **M** | **SD** | **M** | **SD** | |
| Village construction | Increased tourism visibility | 4.21 | 0.743 | 4.26 | 0.691 | 0.865 |
| | Increase the quality of village tourism services and activities | 4.19 | 0.715 | 4.11 | 0.744 | 0.209 |
| | Add travel activity boards or indicators, billboards | 4.09 | 0.736 | 4.11 | 0.750 | 0.334 |
| | Increase the construction of recreational facilities in the village | 4.15 | 0.748 | 4.21 | 0.697 | 0.822 |
| | Promote Yacht Development Organization | 4.13 | 0.713 | 4.25 | 0.644 | 0.546 |
| Quality of life | Facilitate the development of local youth returning to their hometowns | 3.90 | 0.827 | 4.01 | 0.771 | 0.123 |
| | Improve the protection of marine culture | 4.07 | 0.746 | 4.00 | 0.725 | 0.045 * |
| | Increase the protection of coastal style buildings | 3.96 | 0.836 | 3.97 | 0.818 | 0.498 |
| Culture and safety | Harmonious interaction between tourists and residents | 3.96 | 0.867 | 4.06 | 0.813 | 0.760 |
| | Reduce the burden on police, fire and security personnel | 3.79 | 0.876 | 3.86 | 0.844 | 0.273 |
| | Willingness to revisit or buy property on the island | 3.89 | 0.807 | 3.95 | 0.745 | 0.213 |

* $p < 0.05$.

### 4.4. Analysis of the Perceptions of Different Stakeholders on the Environmental Impact of Tourism in Island Villages during the Epidemic

From the literature, it is known that the existing village communities and the surrounding marine environment will be changed to some extent during the promotion of the tourism industry and development [14,16,37]. The analysis showed that, in the community environment, increased littering by tourists (3.82:3.84), insufficient boat-mooring space (3.80:3.83), insufficient tourist resting and parking facilities (3.73:3.70), increased heavy oil pollution (3.69:3.74), and maintenance of landscape and historical sites (3.58:3.72) and in the ecological environment, increased waste on the harbor surface (3.83:3.95), increase in oil pollution on the port surface (3.74:3.77), destruction of marine habitats (3.60:3.76), increase in heavy oil discharge pollution (3.52:3.69), increase in tourism waste (3.64:3.90), and the threat of foreign species (3.30:3.53) showed significant differences ($p < 0.05$), while no significant differences were found in other issues. It can be seen that, on the social level of local tourism development on the island, different stakeholders have different views on the current situation of local boat-mooring space, tourist resting and parking facilities, the effectiveness of landscape and historical sites maintenance, tourists' littering behavior, heavy oil discharge pollution, sea-surface waste and oil pollution on the port, destruction of marine life habitat, heavy oil discharge pollution, tourism waste, and threats from foreign species. Visitors feel more strongly about littering, boat-mooring space, heavy oil discharge pollution, landscape and historical sites protection, marine waste and pollution on the harbor, marine habitat, heavy oil discharge, tourism waste, and threats from foreign species,

while residents feel more strongly about the lack of tourist resting and parking facilities. As shown in Table 6.

**Table 6.** Analysis of the perceived environmental impact of tourism in island villages.

| Secondary Facets | Issues | Resident | | Tourist | | p |
|---|---|---|---|---|---|---|
| | | M | SD | M | SD | |
| Community environment | The problem of tourists littering has increased | 3.82 | 1.030 | 3.84 | 0.828 | 0.002 * |
| | Ship berthing space is still insufficient | 3.80 | 0.861 | 3.83 | 0.709 | 0.001 * |
| | Tourist rest and parking facilities are still insufficient | 3.73 | 0.886 | 3.70 | 0.767 | 0.010 * |
| | Tourists are the cause of damage to the environmental quality of tourist villages | 3.77 | 0.912 | 3.86 | 0.835 | 0.118 |
| | Increase pollution caused by heavy oil emissions | 3.69 | 1.050 | 3.74 | 0.872 | 0.028 * |
| | Increase in the amount of tourism waste | 3.86 | 0.992 | 3.84 | 0.837 | 0.190 |
| | Historic sites restoration | 3.58 | 0.881 | 3.72 | 0.791 | 0.005 * |
| Ecological environment | Increase in garbage in ports and on the sea | 3.83 | 0.955 | 3.95 | 0.772 | 0.014 * |
| | Increased port water pollution | 3.74 | 1.018 | 3.77 | 0.839 | 0.019 * |
| | Excessive development of the coast or land around the port | 3.65 | 0.923 | 3.75 | 0.819 | 0.139 |
| | There are enough public toilets around the port | 3.23 | 0.998 | 3.42 | 0.852 | 0.066 |
| | The destruction of marine habitats | 3.60 | 1.023 | 3.76 | 0.821 | 0.000 * |
| | Marine pollution caused by heavy oil discharge | 3.52 | 1.093 | 3.69 | 0.899 | 0.004 * |
| | Increased tourism garbage | 3.64 | 1.004 | 3.90 | 0.770 | 0.000 * |
| | Alien species threaten local ecology | 3.30 | 1.069 | 3.53 | 0.810 | 0.000 * |

* $p < 0.05$.

## 5. Discussion

### 5.1. Economic Aspects

The analysis shows that different rights holders have different perceptions of job opportunities, job and salary income, and the return of tourism benefits to the community. Residents feel more strongly about changes in job opportunities and wages, while tourists feel more strongly that the benefits of tourism development are clearly paying back to the community, which is consistent with the literature [1,16].

Our research suggests that although the epidemic has brought about a crisis in the global tourism environment and industry, Taiwan's internal control of the epidemic has been effective, and domestic tourism activities have not been affected for the time being. In addition, due to topographical factors, Penghu has few mineral resources and heavy industries and thus few employment opportunities. The current focus is mostly on the tourism service industry, but domestic tourism and consumption alone cannot make up for the lack of economic income generated by foreign tourists. Although the global tourism industry has been hit by the epidemic, the current economic situation in Penghu is still able to meet the operating costs of the local tourism industry and provide stable jobs thanks to the consumption of Taiwanese tourists, but it cannot compare to the tourism boom before the epidemic. Currently, there is no improvement in the scale of local community building in the face of funding shortages. As a result, different stakeholders feel differently about the effectiveness of the current tourism development in the island villages in terms of the impact of the epidemic on local job opportunities, job and salary income, and the return of tourism development benefits to the community. Residents feel more strongly about the changes in job opportunities and income, while tourists feel more strongly about the changes in the effectiveness of the benefits of tourism development in giving back to the community.

### 5.2. Social Aspects

The analysis revealed that different stakeholders felt differently about the effectiveness of local tourism development on the island due to the impact of the epidemic and that

residents had higher perceptions of the effectiveness of marine culture preservation, which is not consistent with the literature [1,37].

The study concluded that although the global tourism environment is affected by the epidemic, Penghu, which originally relied on marine resources for its livelihood, has made use of technology and vessels and other resources to develop its tourism industry in recent years. Due to the topography of the island, the buildable space is limited. In addition, it is one of the national parks in Taiwan, and the development and construction are restricted by relevant laws and regulations. The limited space and scale of local development and construction have resulted in the preservation of coastal architecture and culture in Penghu. However, due to the promotion of tourism development for many years, the local construction has been adjusted according to the needs of tourists, which facilitates tourists to engage in tourism activities but also loses the original architectural appearance and cultural characteristics. As a result, different stakeholders have different perceptions of the effectiveness of the preservation of local marine culture due to the impact of the epidemic on the current tourism development of the island villages, and residents have higher perceptions of the effectiveness of the preservation of marine culture.

*5.3. Environmental Aspects*

The analysis showed that there were significant differences in perceptions of different stakeholders regarding the effectiveness of the current tourism development on the island due to the impact of the epidemic, such as the effectiveness of maintaining local boat-mooring space, tourist resting and parking facilities, landscape and historical sites, the littering behavior of tourists, heavy oil discharge pollution, waste garbage and oil pollution on the harbor surface, destruction of marine habitats, heavy oil discharge pollution, tourism waste, and threats from foreign species. Visitors feel more strongly about littering, boat-mooring space, heavy oil discharge pollution, landscape and historical sites protection, marine waste and pollution on the harbor, marine habitat, heavy oil discharge, tourism waste, and threats from foreign species, while residents feel more strongly about the lack of tourist resting and parking facilities, which is not consistent with the literature [14,15,31].

Our study concluded that the threat of the epidemic is still hovering around the world, resulting in a constrained tourism environment and a certain degree of impact on the operation of the local tourism industry in Penghu. Tourism activities in Taiwan have not been banned, and there are still domestic tourists visiting Penghu to spend money, so the marine tourism activities and vessel operations around the area are still active. Although the number of tourists and the frequency of boat operations in Penghu have decreased instantly under the impact of the epidemic environment, it is difficult to completely remove and clean up the tourism waste and oil pollution left by a large number of tourists and boat operations under the long-term tourism development. In addition, the number of tourists and the frequency of tourism activities have dropped sharply, and the local government and the tourism industry are short of funds and restricted in development and maintenance, which has led to the emergence of problems, such as tourism waste and oil pollution in the village community and the surrounding marine environment.

The local tourism activities and the number of tourists in Penghu have obviously shrunk under the epidemic, but the problem of insufficient parking and open space in Penghu already exists. The problem of epidemic hazards will be effectively controlled in the tourism environment, as the vaccine will be developed to provide effective protection for human beings. Even though the epidemic is affecting the local tourism industry, there is still a need to solve the problem of inadequate tourist rest and parking facilities. As a result, different stakeholders feel differently about the severity of the current tourism development in the island villages due to the impact of the epidemic, such as the effectiveness of local boat-mooring space, tourism resting and parking facilities, maintenance of landscape and historical relics, tourists' littering behavior, heavy oil discharge pollution, harbor surface waste and oil pollution, destruction of marine habitats, pollution of heavy oil spills, tourism waste, and threats from foreign species. Tourists feel more about the increase in littering,

lack of space for boat mooring, increase in heavy oil discharge pollution, increase in the maintenance of historical and scenic sites, increase in oil pollution in the harbor, destruction of marine habitats, increase in heavy oil discharge pollution, increase in tourism waste, and the threat of foreign species, while residents feel more about the lack of tourist rest and parking facilities.

## 6. Conclusions and Recommendations

The global tourism industry has been hit by the COVID-19 epidemic, and so has the island region. The study found that, due to the rich cultural history and marine resources of island villages, local marine culture still has sufficient tourism appeal to reduce the impact and loss of the tourism industry through domestic tourism. However, the shortage of funds in tourism areas will limit the effectiveness of local job opportunities, job income, and the return of tourism development benefits to the community. The negative impact of tourism development on island villages and the surrounding marine environment can be seen in the reduced frequency of tourism activities and vessel operations due to the impact on the main source of funding and the relative reduction in maintenance manpower.

Based on the above results, we suggest the following:

1. For the local government

Although funds for Penghu are limited due to the epidemic, local governments can take advantage of this period of low tourist desire to redesign scenic areas and maintain and enhance their sanitation and safety. With the limited number of tourists, the government should re-plan the tourism development strategy, control the current participants' accommodation, transportation and tourism activities, set the number of people and activity routes in public areas, invest more money to enhance the disinfection and cleaning of the tourism area and the surrounding marine environment, and finally redesign the inadequacy of parking and resting areas in the tourism area and the construction of marine cultural features.

2. For communities and businesses

The epidemic has reduced tourism numbers, affected local tourism businesses, and impacted people's lives. However, for local residents working in the tourism service industry, more time is available for personal household maintenance and cleaning, and even community-development associations can be brought together to clean up the village environment, thus addressing the environmental pollution of the village and marine ecosystem caused by tourism development. Only in this way can we maximize the use of resources, reduce the operating costs of local enterprises, tide over difficult times, and make local industries sustainable.

3. For the follow-up study

The present study uses a mixed methods approach to examine the current state of tourism development on the island during the COVID-19 epidemic environment and the risk of infection. Indeed, the risk of infection environment is an invisible stressor for both residents and tourists that will affect individuals' physical and mental health and desire to travel. Therefore, it is suggested that follow-up studies could be conducted to address relevant aspects to fill the research gap.

**Funding:** This research received no external funding.

**Institutional Review Board Statement:** Not applicable.

**Informed Consent Statement:** Not applicable.

**Data Availability Statement:** No data support.

**Acknowledgments:** I thank everyone who assisted in completing the research.

**Conflicts of Interest:** The authors declare no conflict of interest.

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
