# Peer review of "A Study on the Current Impact on Island Tourism Development under COVID-19 Epidemic Environment and Infection Risk: A Case Study of Penghu"

_sustainability, doi:10.3390/su131910711_

Round 1

Reviewer 1 Report

The manuscript addresses a very interesting issue, relating tourism with territory and the current pandemic crisis. The argument is solid, and methods are well explained and used to provide answer to the major topics of the paper. However, the impact of Covid-19 on the island tourism is not clear on the text, as some background/changes data is missing. Thus, I would suggest the authors to improve the manuscript, regarding:

1) Introduction can have detailed information on the importance of island tourism at Penghu. It would be useful to understand what the main trends/dynamics are (number of tourists, touristic accommodation units, economic impact, …) over the last decade. It would also help understand how Covid-19 has impacted this activity. This brief analysis/presentation should also consider the historic relation of tourism with locals. Has it always been identified as something good for the island? Did the pandemics change the perceptions?

2) Section 2 can introduce one more topic: touristification/overtourism. Is it an issue that affects island tourism?

3) Conclusion should go further on the analysis. The recommendations are too vague. For example, it would be important to understand what the main public policies/planning are and how is local government managing tourism (before and during the pandemics) to support the proposals.

Author Response

1) Introduction can have detailed information on the importance of island tourism at Penghu. It would be useful to understand what the main trends/dynamics are (number of tourists, touristic accommodation units, economic impact, …) over the last decade. It would also help understand how Covid-19 has impacted this activity. This brief analysis/presentation should also consider the historic relation of tourism with locals. Has it always been identified as something good for the island? Did the pandemics change the perceptions?

..

Thank you for your suggestion. We have added relevant information, such as lines 43-59.

2) Section 2 can introduce one more topic: touristification/overtourism. Is it an issue that affects island tourism?

...

Thank you for your suggestion. We have added relevant information, such as lines 107-138.

3) Conclusion should go further on the analysis. The recommendations are too vague. For example, it would be important to understand what the main public policies/planning are and how is local government managing tourism (before and during the pandemics) to support the proposals.

Thank you for your suggestion. We reinforce the content of the recommendations of the government, business units, and development decision-making, and explain the important factors of sustainable development.

Reviewer 2 Report

The paper is very interesting, it deals with a very important and current topic of the impact of the COVID-19 pandemic on island tourism development. The paper has the right structure, the results are elaborated using statistical methods and then discussed. Despite the high value of the paper, following improvements should be applied to, increase the quality of paper:

  1. References to sustainable development are rare and dispersed. I propose to enrich the text with considerations on sustainability (including its dimensions and indicators, the importance of health in sustainable development) and sustainable tourism (including the difference between mass tourism and sustainable tourism).
  2. There is no reference to noise, small repetitions of the content may be seen.
  3. Penghu Island as a case study needs more characterisation, a deeper justification why it was chosen; there should be a figure showing its location, presenting data on tourism traffic (including its structure) and changes related to COVID-19; can it be considered representative?
  4. The title of Table 5 is a repetition of the title of Table 4, but it is about social impact
  5. The discussion should be enriched with references to the work of other authors
  6. A reference to sustainable development in the conclusions is advisable

In addition, it is advisable to proofread the entire text linguistically and to add photographs.

I strongly encourage you to improve your paper.

Author Response

  1. References to sustainable development are rare and dispersed. I propose to enrich the text with considerations on sustainability (including its dimensions and indicators, the importance of health in sustainable development) and sustainable tourism (including the difference between mass tourism and sustainable tourism).

...

Thank you for your suggestion. We have added relevant information, such as lines 107-138.

2.There is no reference to noise, small repetitions of the content may be seen.

Thank you for your suggestion. We have added relevant information, such as lines 53-54.

3.Penghu Island as a case study needs more characterisation, a deeper justification why it was chosen; there should be a figure showing its location, presenting data on tourism traffic (including its structure) and changes related to COVID-19; can it be considered representative?

…..

Thank you for your suggestion. We have added relevant information, such as lines 43-59.

4.The title of Table 5 is a repetition of the title of Table 4, but it is about social impact

Thank you for your suggestion. We have revised the title names of Table 5.

5.The discussion should be enriched with references to the work of other authors

Thank you for your suggestion. In each of our discussion chapters, the first paragraph is for literature review and analysis. We will review and compare the source of the literature after the explanation of this paragraph to strengthen the logic of the discussion.

6.A reference to sustainable development in the conclusions is advisable

--

Thank you for your suggestion. We reinforce the content of the recommendations of the government, business units, and development decision-making, and explain the important factors of sustainable development.